# The Impact of AI on Metal Artifacts in CBCT Oral Cavity Imaging

**DOI:** 10.3390/diagnostics14121280

**Published:** 2024-06-17

**Authors:** Róża Wajer, Adrian Wajer, Natalia Kazimierczak, Justyna Wilamowska, Zbigniew Serafin

**Affiliations:** 1Department of Radiology and Diagnostic Imaging, University Hospital No. 1 in Bydgoszcz, Marii Skłodowskiej—Curie 9, 85-094 Bydgoszcz, Poland; justyna.wilamowska@cm.umk.pl (J.W.); serafin@cm.umk.pl (Z.S.); 2Independent Resercher, 86-005 Zielonka, Poland; adrianwajer1@gmail.com; 3Kazimierczak Private Medical Practice, Dworcowa 13/u6a, 85-009 Bydgoszcz, Poland; natnowicka@gmail.com; 4Department of Radiology and Diagnostic Imaging, Collegium Medicum, Nicolaus Copernicus University in Torun, Jagiellońska 13-15, 85-067 Bydgoszcz, Poland

**Keywords:** cone-beam computed tomography, deep learning model, image quality, noise reduction, dental imaging, metal artifact reduction

## Abstract

Objective: This study aimed to assess the impact of artificial intelligence (AI)-driven noise reduction algorithms on metal artifacts and image quality parameters in cone-beam computed tomography (CBCT) images of the oral cavity. Materials and Methods: This retrospective study included 70 patients, 61 of whom were analyzed after excluding those with severe motion artifacts. CBCT scans, performed using a Hyperion X9 PRO 13 × 10 CBCT machine, included images with dental implants, amalgam fillings, orthodontic appliances, root canal fillings, and crowns. Images were processed with the ClariCT.AI deep learning model (DLM) for noise reduction. Objective image quality was assessed using metrics such as the differentiation between voxel values (ΔVVs), the artifact index (AIx), and the contrast-to-noise ratio (CNR). Subjective assessments were performed by two experienced readers, who rated overall image quality and artifact intensity on predefined scales. Results: Compared with native images, DLM reconstructions significantly reduced the AIx and increased the CNR (*p* < 0.001), indicating improved image clarity and artifact reduction. Subjective assessments also favored DLM images, with higher ratings for overall image quality and lower artifact intensity (*p* < 0.001). However, the ΔVV values were similar between the native and DLM images, indicating that while the DLM reduced noise, it maintained the overall density distribution. Orthodontic appliances produced the most pronounced artifacts, while implants generated the least. Conclusions: AI-based noise reduction using ClariCT.AI significantly enhances CBCT image quality by reducing noise and metal artifacts, thereby improving diagnostic accuracy and treatment planning. Further research with larger, multicenter cohorts is recommended to validate these findings.

## 1. Introduction

Since its wider commercial introduction in the 2000s, cone-beam computed tomography (CBCT) has become an invaluable tool in dental imaging, offering high-resolution, three-dimensional representations of the oral cavity [1,2]. It allows for precise, submillimeter assessments that are particularly useful in endodontics, implant procedure planning, and cephalometry [3]. However, the presence of metallic dental objects, such as amalgam fillings, crowns, orthodontic appliances, root fillings and implants, often leads to significant image artifacts of varying appearance and extent, complicating accurate diagnosis and treatment planning [4].

CBCT artifacts are produced by discrepancies between mathematical models and actual imaging processes [5]. Beam hardening is the most significant metal-induced artifact of CBCT and is one of the main factors responsible for diminished image quality of the entire CBCT examination [5,6]. Closer to dental foreign materials, beam hardening and noise induce excessive gray value variation, obscuring critical anatomical structures and compromising diagnostic accuracy [7,8]. Image noise manifests as a disturbance in a signal, and reduced resolution at low contrast and can significantly impair the quality of CBCT images [9]. Moreover, noise and scatter are important factors influencing the creation of new artifacts [5].

Traditional methods for metal artifact reduction involve manual correction or postprocessing techniques—metal artifact reduction (MAR) algorithms. The first MAR technique includes projection data inpainting, which was introduced in the late 1980s [10]. Newer, more sophisticated groups of MAR solutions are the iterative reconstruction (IR) algorithms, which have proven effective in numerous applications, including CBCT [11,12]. Recently, convolutional neural networks (CNNs) have been explored as a potential solution for reducing metal artifacts. Preliminary studies have already shown their effectiveness in artifact reduction in both CT and CBCT [13,14].

Recently, artificial intelligence (AI) has attracted significant interest in the field of dentistry, particularly in orthodontics and dentomaxillofacial imaging [15,16,17]. Recent years have led to the development of deep learning-based image reconstruction algorithms (DLRs) that can effectively reduce excess image noise [18]. Initially, three CT vendors and two independent (vendor-agnostic) software companies released deep learning (DL) algorithms approved by the U.S. Food and Drug Administration: the first one, TrueFidelity (GE Healthcare, Wauwatosa, WI, USA), in 2019; the second, AiCE (Canon Medical Systems, Tustin, CA, USA); the latter, Precise Image (Philips Healthcare, Amsterdam, The Netherlands); and vendor-agnostic software companies, ClariCT.AI (ClariPi) and PixelShine (AlgoMedica, Heidelberg Germany) [18]. One of the available vendor-neutral deep learning models (DLMs) is ClariCT (ClariPi, Seoul, South Korea), which operates in the image postprocessing stage. According to the vendor, the algorithm was trained on a dataset of over a million CT images from various CT systems and reconstruction settings [19]. ClariCT.AI is based on a modified U-net-type convolutional neural network (CNN) model and has already been proven to reduce image noise and provide high diagnostic accuracy [20,21,22,23,24,25]. ClariCT.AI has already been proven to be able to maintain image quality while offering up to a 70% dose reduction [19,26].

Hypothetically, this AI denoising tool could also positively impact the quality parameters of CBCT images by reducing additional noise associated with metal artifacts. To the best of our knowledge, no studies have yet analyzed the application of DLM in dental CBCT, along with an objective and subjective assessment of image quality parameters of metal artifact reduction.

The aim of this study was to assess the impact of AI-driven noise reduction algorithms on metal artifacts and image quality parameters in CBCT images of the oral cavity.

## 2. Materials and Methods

This retrospective, anonymized study was approved by the local institutional review board, which waived the requirement for informed consent (decision reference No. KB 227/2023).

### 2.1. Population and Sample Size Calculations

The study material initially involved 70 patients (27 men and 43 women, aged 15–71). All CBCT scans were acquired at a single private dental center. All patients were referred for CBCT scans by orthodontists and dental surgeons between January and September 2023. The indications for CBCT imaging were the presence of impacted teeth and the suspicion of periapical lesion appearance.

The main study inclusion criteria were as follows: images obtained with a standard radiation dose protocol and the presence of intraoral metal dental objects causing noticeable artifacts. Patients whose images were burdened with severe motion artifacts were excluded from the study.

To determine the required sample size for detecting a significant difference in the contrast-to-noise levels (CNRs) between DLM and native reconstructions in cone-beam computed tomography (CBCT) images, we performed a paired *t* test. The following assumptions were made: significance level (α): 0.05, power (1-β): 0.80.

### 2.2. CBCT Scanning Protocol and Image Reconstruction

All scans were conducted using a Hyperion X9 PRO 13 × 10 CBCT machine (MyRay, Bologna, Italy). A single, standard configuration referred to as the “Regular” setting was utilized, characterized by 90 kV, 36 mAs, CTDI/Vol 4.09 mGy, and a 13 cm field of view. The images were reconstructed with a slice thickness of 0.3 mm. Following the scanning process, the images were anonymized and exported for further analysis. The denoised reconstructions were achieved using the commercially available DLM software ClariCT.AI. The scheme of the study design is presented in Figure 1.

### 2.3. Objective Image Quality Assessment

Image quality analyses were performed on a dedicated workstation using the open-source software ImageJ version 1.53t (National Institutes of Health, Bethesda, MD, USA) [27]. Objective analysis was performed by a radiologist with 3 years of experience in craniofacial CT assessment. The rectangular regions of interest (ROIs) were placed at two separate axial slices for every analyzed object—one with the most pronounced artifacts and the other closest slices without artifacts. Five ROIs were placed on each slice:-ROIs 1 and 2 in the near field (<10 mm) of the metal object, anterior and posterior to the object, preferable to homogenous bone tissue (spongious bone, dentine);-ROI 3 in the air anterior to the skin;-ROI 4 within the tongue musculature;-ROI 5 within the subcutaneous adipose tissue of the cheek closest to the metal object.

ROIs were placed in the areas of the most pronounced artifacts (first slice) and then carefully manually adjusted to cover the same type of homogenous tissue on the second, nonartifact slice. To avoid misregistration, the ROIs were automatically propagated to both types of reconstructions (native and DLM). Each ROI contained the mean gray density voxel value (VV) and standard deviation (SD) of the VV. The sample ROI positions are shown in Figure 2.

The image quality assessment covered each of the following dental objects: dental implants, amalgam fillings, orthodontic appliances, root canal fillings, and crowns. The specific information of the metallic material was not retrieved from the patients.

To assess the objective image quality parameters and MAR properties of the DLM reconstructions, the ΔVV AIx and CNR were calculated. The ΔVV and AIx were calculated with previously described formulas [28]. The ΔVV was calculated to objectively assess the intensity of artifacts by measuring the VV at the worst artifacts and in the corresponding control, artifact-free ROI. In detail, ΔVV was calculated as follows:

Formula (1): The differentiation between the VVs:ΔVV = | VV_max artifact_ − VV_control_ |(1)

The VV_max artifact_ is the most pronounced artifact in the near field, and VV_control_ is the measurement in the closest axial slice without artifacts in the corresponding homogenous tissue (dentine, spongious bone, etc.).

To quantify the severity of the artifacts, the AIx was calculated using Formula (2):

Formula (2): Artifact index:(2)$$\mathrm{AIx}=\sqrt{{SD1}^{2}-{SD2}^{2}}$$
where *SD*1 is the standard deviation in the ROI with the most artifacts and *SD*2 is the standard deviation in the control ROI in the region without artifacts.

The contrast-to-noise ratio (CNR) was calculated with the formula presented by Fontenele et al. [29].

Formula (3): Contrast-to-noise ratio:(3)$$\mathrm{CNR}=\frac{\left|VVmax\;artifact-VVcontrol\right|}{\sqrt{{SDmax\;artifact}^{2}+{SDcontrol}^{2}}}$$

The ΔVV, AI and CNR values of the appropriate ROIs in native and DLM images were compared to evaluate the effectiveness of the AI denoising tool.

### 2.4. Subjective Image Quality Assessment

Subjective image quality was evaluated by a radiologist and a dentist, both of whom had over 5 years of experience in craniofacial CT assessment. The readers were blinded to patient information and whether an AI denoising tool had been used. The evaluation was carried out twice at one-month intervals. During these assessments, two parameters were assessed: the overall image quality and the intensity of the artifacts. The readers rated the overall image quality with a previously described 5-point scale (1 = poor, 5 = excellent) [25]. Image noise, sharpness, and the visibility of anatomical structures were considered:

5—Excellent delineation of structures and excellent image quality;

4—Clear delineation of structures and good image quality;

3—Anatomical structures still fully assessable in all parts and of acceptable image quality;

2—Structures identifiable with adequate image quality;

1—Anatomical structures not identifiable, images with no diagnostic value.

The artifact intensity was assessed within the near field from the metal object (<15 mm) with the use of the following 4-point scale:

1 = Absent/None;

2 = Weak;

3 = Medium;

4 = Strong.

The assessments were performed only on slices containing metal objects. Figure 3 presents samples of artifact intensity ratings.

### 2.5. Inter- and Intrareader Agreement

To assess inter-reader agreement, the results of subjective image analysis were compared with the results of the intraclass correlation coefficient (ICC).

To assess intrareader agreement, the results of the first and repeated subjective image analyses were compared.

### 2.6. Statistical Analysis

The means, standard deviations, medians, quartiles, and ranges of the quantitative variables were calculated. The paired Wilcoxon test, Kruskal–Wallis test and post hoc Dunn test were used to compare two repeated measures of quantitative variables. The inter-rater reliability of the qualitative measures between two raters was assessed with the ICC2. The significance level was set to 0.05. All analyses were conducted using R software version 4.3.2.

## 3. Results

### 3.1. Patient Population and Sample Size

The authors reviewed the CBCT scans of 70 patients. Nine patients were excluded due to severe motion artifacts and a scan range not covering the periapical areas of all the present teeth. Finally, the CBCT scans of 61 patients were included.

The mean age of all participants was 46.53 years (SD 14.26; median 48; range 18–72). There were 24 males with a mean age of 49.19 years (SD 13.82; range 25–72) and 37 females with a mean age of 44.97 years (SD 14.47; range 18–71).

The analyzed study material consisted of CBCT scans containing 14 dental implants, 5 amalgam fillings, 6 orthodontic appliances, 25 root canal fillings, and 11 crowns.

The sample size calculation revealed that a sample size of 19 patients was sufficient to detect a significant difference in the CNR between DLM and native reconstructions in CBCT images, with a significance level of 0.05 and a power of 0.80. This sample size ensures that the study can detect meaningful differences in image quality due to the application of the DLM algorithm.

### 3.2. Objective Image Quality

The detailed results of the objective image quality assessments are summarized in Table 1 and Table 2.

The average AIx was significantly greater in the native images than in the DLM reconstructions (*p* < 0.001) when all the ROIs were evaluated. The mean AIx in all the ROIs in the native images was 166.65, while that in the DLM images was 158.31. However, when only the near-field ROIs (ROIs 1 and 2) were evaluated, the differences in the Aix between the native and DLM reconstructions were not statistically significant.

The comparison of CNR levels showed a statistically significant difference (*p* < 0.001) for all examined ROI locations, with higher CNR levels observed in the DLM reconstructions. The mean CNR in all the ROIs in the DLM images was 0.93, while in the case of the native reconstructions, it was 0.79. However, the measurements conducted in the near field (ROIs 1 and 2) exhibited lower differences in the mean CNR than did those in all ROIs (ROIs 1–5) combined. The mean CNR in ROIs 1–2 in DLM images was 0,72, while in the case of native reconstructions, it was 0,7. The difference was statistically significant.

Surprisingly, the ΔVV values were similar in both reconstructions; slightly lower ΔVV values were calculated in ROIs very close to the metallic objects, but these differences were not statistically significant. Metallic artifacts were most pronounced with orthodontic appliances, followed by crowns and amalgam fillings, with the weakest artifacts occurring with implants (Table 3). The visual comparison of two types of reconstructions is presented in Figure 4.

### 3.3. Subjective Image Quality

In summary, both readers indicated that the overall image quality rating was significantly greater for DLM reconstructions than for native reconstructions. Moreover, the results of the assessments of artifact intensity also showed significantly (*p* < 0.001) lower values in DLM images. Table 4 presents the summarized results of the subjective image quality assessment.

### 3.4. Intrareader and Inter-Reader Reliability

The ICCs for the subjective image quality analyses were calculated to assess the intrarater agreement and showed fair to excellent agreement among the readers. These results are shown in Table 5.

The inter-reader agreement for the subjective image quality, expressed as the intraclass correlation coefficient (ICC), showed fair to good agreement (Table 6).

## 4. Discussion

The aim of this study was to investigate the ability of AI software to reduce the presence of additional noise and artifacts associated with various dental and metal artifacts in the oral cavity. Our research indicates that AI can reduce the intensity of metallic artifacts by improving objective parameters such as VV, CNR and AI, as well as subjective parameters such as artifact intensity and overall image quality. The results of our research demonstrate the significant potential of evaluated AI technology in mitigating the effects of metallic artifacts in CBCT imaging.

The current state-of-the-art MAR includes a variety of techniques, each with specific strengths. Projection-based MAR algorithms and dual-energy CT (DECT) are widely used due to their effectiveness in reducing photon starvation and beam-hardening artifacts. The use of high-photon-energy virtual monoenergetic images (VMIs) has proven effective in various applications [30,31,32]. There are initial, positive reports showing the application of these techniques in CBCT imaging [28,33]. Other approaches, such as iterative deblurring methods and segmentation-based approaches, offer significant improvements in image quality [34,35]. However, some authors have shown that their use may hinder some crucial anatomical details and negatively affect the diagnostic accuracy of the examination [36,37,38,39]. The advent of AI in MAR techniques has led to the possibility of combining DL and traditional MAR algorithms. An important study by Gjesteby et al. [40] showed that this combination significantly improved metal artifact reduction in CT images, potentially enabling more accurate tumor volume estimation for radiation therapy planning. In our opinion, this research shows a very promising direction for future research, which may evaluate its application in CBCT imaging.

Our study showed that DLM reconstructions significantly reduced the AIx compared to that of native images when all ROIs were evaluated. This reduction in AIx suggested that the evaluated DLM effectively mitigated the intensity of the metal artifacts. However, compared to the results of the evaluation narrowed to the ROIs in the near field of the metal foreign bodies, the differences in the AIx across both reconstructions were not statistically significant. Interestingly, the ΔVV did not significantly differ between native and DLM images, suggesting that while the DLM reduces additional image noise, it maintains the overall density distribution of the image. In our opinion, these results clearly show that the evaluated AI model has very limited potential for reducing near-field metal artifacts; however, it significantly improves the image quality properties in the far field. This issue stems from the main application and advantages of ClariCT.AI—the noise reduction. To date, no studies have evaluated the impact of the selected DLM on metal artifacts. However, a growing body of evidence shows that DL–based MAR algorithms have the potential to remove metal artifacts more accurately than current state-of-the-art MAR methods [18]. A recent study by Park et al. [41] showed that DL methods, such as fidelity-embedded learning, significantly reduce metal artifacts while preserving morphological structures near metallic objects in dental CBCT. In a recent article from 2024, Song et al. [42] proposed a bidirectional artifact representation learning framework to adaptively encode metal artifacts. The authors proved that their method showed superior performance over existing methods in restoring the structural integrity of dental tissues and removing artifacts effectively. Rohleder et al. evaluated the efficacy of cross-domain neural network approaches for metal segmentation in CBCT images acquired with a C-arm [43]. The authors have shown the high accuracy and efficiency of the evaluated algorithms, which outperform traditional threshold-based methods. Hu et al. [44] utilized an AI model with the Wasserstein loss function (WGAN) to effectively reduce noise and artifacts in low-dose dental CTs, outperforming other methods such as general GANs and CNNs.

Our findings showed that the CNR values were significantly greater in DLM images, indicating improved image clarity and better differentiation of anatomical structures. These results are consistent with those of a recent study in which the authors reported similar results for the CNR values of DLM-reconstructed and native CBCT images of temporomandibular joints [24]. Nevertheless, the literature concerning noise optimization in dentomaxillofacial CBCT examinations is very limited. In their 2023 CBCT study, Ramage et al. [45] investigated the impact of standard filtered back projection (FBP) and IR on image noise. The researchers found that IR significantly reduced image noise compared to FBP images, with mean noise levels two times greater for FBP. Other studies [46,47] have explored the effectiveness of generative AI in minimizing image noise and in accessing dentomaxillofacial CT images. Hegazy et al. (2020) [46] improved low-dose dental CT image quality using a WGAN, despite oversmoothing of the small anatomical details. Their 2021 study [47] revealed that variations in WGANs enhanced image quality and reduced noise in half-scan CTs.

All of the abovementioned results are consistent with previous studies that have highlighted the effectiveness of DL algorithms in reducing image noise and enhancing diagnostic accuracy in CT and CBCT imaging [25,48,49,50,51]. The improved CNR and reduced AIx observed in our study further support the potential of DLMs in clinical practice. Partial deep learning-based approaches are highly effective in optimizing CBCT by reducing noise and improving image quality. These methods not only enhance the visual quality of CBCT images but also offer efficient solutions for real-time applications. The integration of advanced DL architectures, such as GANs and U-nets, along with adaptive noise and MAR techniques, holds great potential for further advancements in CBCT imaging.

Subjective assessments by experienced radiologists and dentists revealed a preference for DLM images over native images. The overall image quality was rated significantly higher for DLM reconstructions, with better delineation of anatomical structures and reduced artifact intensity. This finding aligns with previous abovementioned research indicating that AI-based MAR techniques can enhance the visual quality of CBCT scans, making it easier for clinicians to interpret the images accurately. However, although the subjective image quality assessment showed significant improvement in artifact perception, this improvement stems from the lower noise levels rather than from actual artifact reduction. Although the results of subjective artifact intensity assessments showed significant differences between native and DLM reconstructions, the difference was small (3.22 vs. 3.55, respectively). The improved subjective image quality and reduced artifact intensity are crucial for clinical applications, as they can directly influence diagnostic confidence and treatment planning. For example, the reduced artifacts around dental implants and root canal fillings facilitate better visualization of the surrounding bone structures, which is essential for assessing implant integration and detecting potential complications.

This study also examined the impact of various metallic objects on image quality. Orthodontic appliances were found to produce the most pronounced artifacts, followed by crowns and amalgam fillings. Interestingly, implants generated the least severe artifacts. These findings show that the type of metallic object plays a significant role in artifact formation and that DLMs can differentially mitigate these artifacts. Our findings are in line with other studies showing similar results [4,52,53]. This differential impact highlights the importance of tailored MAR strategies depending on the clinical scenario and the types of metallic objects present.

Further research is also warranted to explore the potential of DLMs for reducing radiation doses without compromising image quality. Given the promising results of dose reduction in CT imaging with AI-based reconstruction algorithms [19,26], similar benefits might be achievable in CBCT imaging.

Although the results are promising, this study has several limitations. The retrospective design and relatively small sample size may limit the generalizability of the findings. Future studies should include larger, multicenter cohorts to validate these results. Additionally, the study focused on a single DLM algorithm (ClariCT.AI), and comparative studies with other AI-based noise optimization and MAR techniques would provide a more comprehensive understanding of the effectiveness of various DLMs. One more limitation is the lack of precise knowledge about the composition of the tested metal objects. Future research needs to analyze whether these compositions, sizes and shapes of given metallic bodies influence the formation of artifacts in CBCT imaging and how artificial intelligence can be adapted to meet these specific challenges. Another significant limitation is the restriction of the study material to one CBCT machine and its specific apparatus settings. Utilizing different apparatus settings (tube current, voltage, field of view) would likely yield different results in image quality assessments. Moreover, the subjective nature of image quality assessment, even for experienced readers, is based on their knowledge and perceptions; hence, assessments among readers may vary significantly. Additionally, we emphasize that our study has a relatively small sample size of 61 patients and employs a retrospective design. Future studies with more extensive multicenter cohorts should be conducted to validate the findings and enhance the generalizability of the results.

## 5. Conclusions

In conclusion, this study demonstrated that DLM reconstructions using ClariCT.AI significantly enhance the objective and subjective image quality of CBCT scans by reducing the image noise associated with the presence of metallic artifacts. These improvements have important clinical implications, potentially leading to more accurate diagnoses and better-informed treatment plans. Further research is warranted to systematize the potential impact of AI on the effectiveness of minimizing metallic artifacts in CBCT and its utility in clinical practice in a wider context.

## Figures and Tables

**Figure 1 diagnostics-14-01280-f001:**
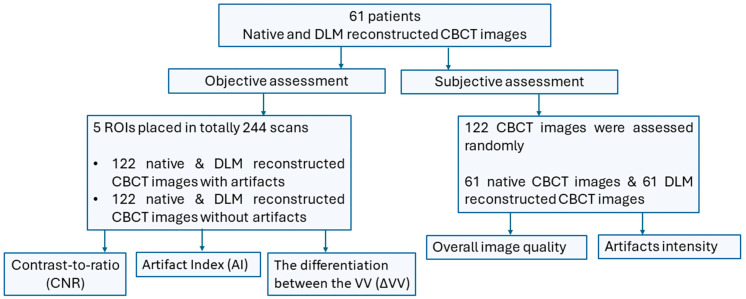
Study design.

**Figure 2 diagnostics-14-01280-f002:**
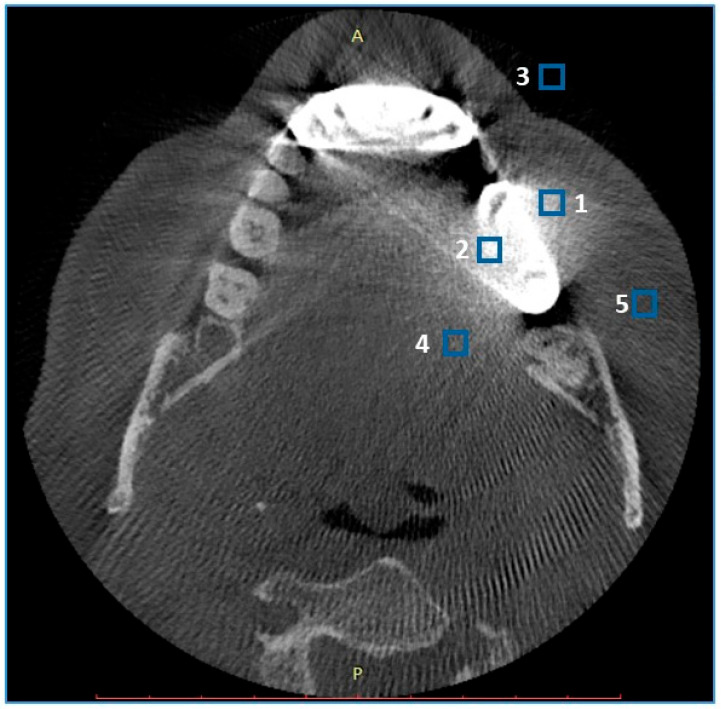
Sample ROI positioning in artifact evaluation. Patients with artifacts due to metal crowns. ROIs: 1—near field; 2—near field; 3—air anterior to the skin; 4—tongue musculature; 5—subcutaneous adipose tissue of the cheek. A—anterior, P—posterior.

**Figure 3 diagnostics-14-01280-f003:**
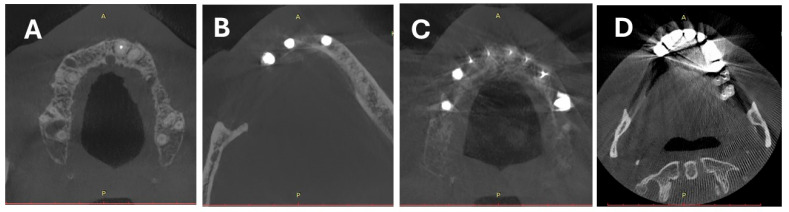
Four-point scale of artifact intensity: (**A**)—absent/none, (**B**)—weak, (**C**)—medium, (**D**)—strong.

**Figure 4 diagnostics-14-01280-f004:**
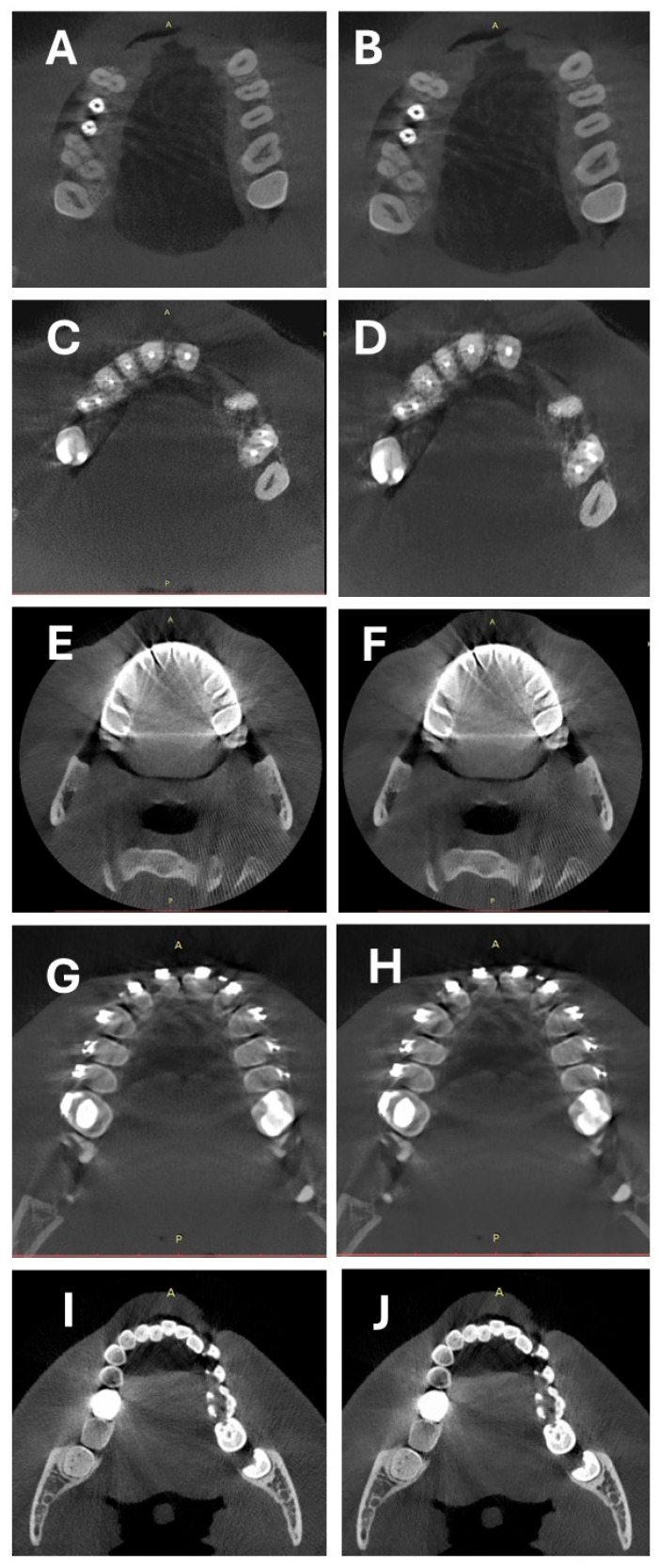
Comparison of two types of reconstructions: (**A**,**B**) artifacts caused by implants, (**C**,**D**) artifacts caused by root canal fillings, (**E**,**F**) artifacts caused by crowns, (**G**,**H**) orthodontic appliances, and (**I**,**J**) artifacts caused by amalgam fillings. (**A**,**C**,**E**,**G**,**I**)—native, (**B**,**D**,**F**,**H**,**J**)—DLM.

**Table 1 diagnostics-14-01280-t001:** Calculations based on all ROI values (ROIs 1–5).

Parameter	Measurement	Mean	SD	Median	Min	Max	Q1	Q3	*p*
ΔVV	Native	174.41	266.69	73.25	0.0	1534.30	27.75	178.62	*p* = 0.094
	DLM	174.07	263.69	71.75	0.1	1506.00	30.47	177.12	
AIx	Native	166.65	218.96	73.89	0.0	1426.54	27.05	204.14	*p* = 0.001 *
	DLM	158.31	199.10	68.23	0.0	1215.26	25.48	205.84	
CNR	Native	0.79	1.05	0.52	0.0	14.13	0.23	1.07	*p* < 0.001 *
	DLM	0.93	1.52	0.61	0.0	23.11	0.26	1.15	

ΔVV—the differentiation between the mean density voxel values, AIx—artifact index, CNR—contrast-to-noise ratio, DLM—deep learning model, SD—standard deviation, Q1—lower quartile, Q3—upper quartile, *p*—Wilcoxon paired test, * statistically significant (*p* < 0.05).

**Table 2 diagnostics-14-01280-t002:** The calculations were based on the values of the ROIs placed in the near field of the artifacts (ROIs 1 and 2).

Parameter	Measurement	Mean	SD	Median	Min	Max	Q1	Q3	*p*
ΔVV	Native	339.64	341.88	205.95	3.60	1534.30	104.20	462.15	*p* = 0.785
	DLM	341.04	335.61	214.25	7.90	1506.00	106.07	449.28	
AIx	Native	379.50	342.29	280.71	0.00	2374.64	159.29	491.80	*p* = 0.214
	DLM	350.92	254.12	281.25	35.19	1301.66	170.38	469.82	
CNR	Native	0.70	0.62	0.54	0.00	3.83	0.23	1.01	*p* = 0.001 *
	DLM	0.72	0.63	0.55	0.01	3.72	0.25	1.04	

ΔVV—the differentiation between the mean density voxel value, AIx—artifact index, CNR—contrast-to-noise ratio, DLM—deep learning model, SD—standard deviation, Q1—lower quartile, Q3—upper quartile, *p*—Wilcoxon paired test, * statistically significant (*p* < 0.05).

**Table 3 diagnostics-14-01280-t003:** Comparison of the impact of different metallic objects on objective image quality parameters.

Parameter	Type of Object	Reconstruction	Mean	SD	Median	Min	Max	Q1	Q3	*p*
ΔVV	Implants	Native	119.70	164.77	53.30	4.60	919.40	23.98	135.80	*p* = 0.125
		DLM	121.27	165.82	53.60	2.10	939.00	26.15	138.43	
	Amalgam fillings	Native	215.32	276.75	108.90	9.40	1231.40	49.00	274.40	*p* = 0.699
		DLM	213.79	281.48	105.90	7.90	1233.30	46.30	314.50	
	Orthodontic appliances	Native	224.82	352.62	70.25	0.30	1505.00	24.45	222.58	*p* = 0.567
		DLM	230.54	352.02	68.50	0.20	1506.00	33.37	357.38	
	Root canal fillings	Native	161.16	212.14	82.75	1.10	1177.90	29.70	186.93	*p* = 0.356
		DLM	160.54	209.45	85.25	0.90	1177.50	31.80	185.95	
	Crown	Native	227.84	389.47	64.00	0.00	1534.30	15.90	160.25	*p* = 0.195
		DLM	222.89	379.95	63.70	0.10	1486.80	22.60	157.65	
CNR	Implants	Native	0.97	1.72	0.60	0.05	14.13	0.26	1.15	*p* = 0.350
		DLM	1.22	2.76	0.74	0.03	23.11	0.31	1.33	
	Amalgam fillings	Native	0.81	0.65	0.62	0.01	2.87	0.42	1.03	*p* = 0.156
		DLM	0.90	0.80	0.64	0.01	3.44	0.38	1.12	
	Orthodontic appliances	Native	0.60	0.77	0.38	0.00	3.83	0.08	0.71	*p* = 0.656
		DLM	0.64	0.76	0.44	0.00	3.72	0.13	0.74	
	Root canal fillings	Native	0.74	0.69	0.52	0.02	4.59	0.28	1.03	*p* = 0.003 *
		DLM	0.85	0.80	0.60	0.01	4.47	0.31	1.11	
	Crown	Native	0.78	0.84	0.41	0.00	3.82	0.18	1.14	*p* = 0.097
		DLM	0.89	0.99	0.55	0.02	4.24	0.18	1.21	
AIx	Implants	Native	105.00	172.18	34.19	5.17	852.14	17.23	133.07	*p* = 0.641
		DLM	105.73	168.95	39.54	2.64	842.48	18.77	128.38	
	Amalgam fillings	Native	178.87	165.61	146.87	8.01	570.81	59.14	233.52	*p* = 0.474
		DLM	168.42	165.18	128.50	0.00	550.87	43.39	222.61	
	Orthodontic appliances	Native	285.51	355.50	138.01	20.56	1426.54	61.63	338.19	*p* = 0.157
		DLM	242.02	282.87	123.68	9.54	1215.26	47.12	356.00	
	Root canal fillings	Native	167.04	199.54	65.43	0.00	996.37	30.91	221.97	*p* = 0.155
		DLM	162.46	191.93	71.11	4.14	1025.77	30.29	234.94	
	Crown	Native	173.85	217.91	103.15	0.57	972.66	25.05	204.70	*p* = 0.247
		DLM	165.63	198.34	85.15	1.31	796.04	25.27	204.76	

ΔVV—the differentiation between the mean density voxel values, AIx—artifact index, CNR—contrast-to-noise ratio, DLM—deep learning model, *p*—Kruskal–Wallis test + post hoc analysis (Dunn test), SD—standard deviation, Q1—lower quartile, Q3—upper quartile, * statistically significant (*p* < 0.05).

**Table 4 diagnostics-14-01280-t004:** The summarized results of the subjective image quality assessment.

Parameter	Measurement	N	Mean	SD	Median	Min	Max	Q1	Q3	*p*
Overall image quality	Native	244	3.16	0.63	3	2	5	3	4	*p* < 0.001 *
	DLM	244	3.86	0.82	4	2	5	3	4	
Intensity of artifacts	Native	244	3.55	0.67	4	1	4	3	4	*p* < 0.001 *
	DLM	244	3.22	0.76	3	1	4	3	4	

DLM—deep learning model, N—number, SD—standard deviation, Q1—lower quartile, Q3—upper quartile, *p*—Wilcoxon paired test, * statistically significant (*p* < 0.05).

**Table 5 diagnostics-14-01280-t005:** Intrareader agreement on subjective image quality parameters.

Reconstruction	Parameter	ICC	95% CI		Agreement (Cicchetti)
Native	Overall image quality	0.515	0.371	0.634	Fair
	Intensity of artifacts	0.757	0.670	0.824	Excellent
DLM	Overall image quality	0.515	0.372	0.634	Fair
	Intensity of artifacts	0.681	0.572	0.767	Good

DLM—deep learning model, ICC—intraclass correlation coefficient, 95% CI—95% confidence interval.

**Table 6 diagnostics-14-01280-t006:** Inter-reader agreement on subjective image quality parameters.

Reconstruction	Parameter	ICC	95% CI		Agreement (Cicchetti)
Native	Overall image quality	0.477	0.326	0.604	Fair
	Intensity of artifacts	0.721	0.624	0.797	Good
DLM	Overall image quality	0.478	0.330	0.604	Fair
	Intensity of artifacts	0.573	0.429	0.686	Fair

DLM—deep learning model, ICC—intraclass correlation coefficient, 95% CI—95% confidence interval.

## Data Availability

Data available upon request.

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
