# Peer review of "The Impact of AI on Metal Artifacts in CBCT Oral Cavity Imaging"

_diagnostics, 2024, doi:10.3390/diagnostics14121280_

Round 1

Reviewer 1 Report

Comments and Suggestions for Authors

The methods section is detailed and thoroughly describes the study protocol, including patient inclusion and exclusion criteria, CBCT scanning parameters, and image quality assessment methods. The use of both objective and subjective image quality assessment methods strengthens the credibility of the findings. However, the methodology could be more transparent by including a diagram or table summarizing all steps of the study. The results are presented clearly and logically, with appropriate use of tables and figures to facilitate understanding of the data. Objective image quality assessments showed significant reductions in metal artifacts and improved contrast-to-noise ratio (CNR) in DLM-processed images. Subjective image quality assessments supported these findings. While the results section is well-prepared, it could benefit from more visual examples of images before and after applying the DLM algorithm.The discussion critically addresses the results, comparing them with the existing literature and discussing potential limitations of the study. The authors appropriately emphasize the need for further research with larger, multicenter cohorts to validate their findings. However, the discussion could better address the potential clinical implications of using DLM in everyday practice and the costs and accessibility of this technology.The conclusions are consistent with the presented results and highlight the potential clinical benefits of using DLM for reducing metal artifacts in CBCT. The authors suggest further research, which is reasonable given the preliminary nature of their findings.

"The Impact of AI on Metal Artifacts in CBCT Oral Cavity Imaging" is a well-conducted study that makes a significant contribution to the field of diagnostic imaging. Despite some minor shortcomings, the work deserves publication in a scientific journal, provided that the suggestions for improving the clarity of the methodology and expanding the discussion on practical clinical aspects are considered.

Author Response

Answers included in the attached cover letter.

Reviewer 2 Report

Comments and Suggestions for Authors

1.      The study excluded patients with severe motion artefacts, potentially limiting the understanding of AI's effectiveness in real-world scenarios where motion artefacts are common.

2.      The study does not retrieve specific information about the metallic materials in dental objects, which could influence artefact formation. The authors should collect detailed information about the types of metals used in dental restorations and implants to understand better their impact on artefact formation and the effectiveness of AI in mitigating these artefacts.

3.      The AI algorithm shows limited effectiveness in reducing artefacts near metallic objects, which is crucial for accurate diagnosis and treatment planning. AI algorithms should be explicitly improved for near-field artefact reduction. Research combining AI with traditional MAR techniques might yield better results for artefacts close to metal objects.

4.      The authors need to compare their results with benchmarks based on traditional MAR techniques to determine the most effective methods for artefact reduction in CBCT imaging.

5.      The study finds varying levels of artefact production from different metallic objects but does not delve deeply into the specific challenges posed by each type. Detailed analyses of how various types of metallic objects (e.g., other metals, sizes, and shapes) affect CBCT imaging and how AI can be tailored to address these specific challenges are needed.

6.      The study has a relatively small sample size of 61 patients and employs a retrospective design. The reviewer suggests conducting future studies with more extensive multicenter cohorts to validate the findings and enhance the generalizability of the results. This statement must be added at the end of the discussion for further investigation

Comments on the Quality of English Language

Minor editing of English language required

Author Response

Answers included in the attached cover letter
